# Compounds without borders: A mechanism for quantifying complex odors and responses to scent-pollution in bumblebees

**Jordanna D. H. Sprayberry** *

Departments of Biology & Neuroscience, Muhlenberg College, Allentown, Pennsylvania, United States of America

* jordannasprayberry@muhlenberg.edu

**Data Availability Statement:** All data in this manuscript are available for download in the supplemental materials.

## Abstract

Bumblebees are critical pollinators whose populations have been experiencing troubling declines over the past several decades. Successful foraging improves colony fitness, thus understanding how anthropogenic influences modulate foraging behavior may aid conservation efforts. Odor pollution can have negative impacts on bumble- and honey-bees foraging behavior. However, given the vast array of potential scent contaminants, individually testing pollutants is an ineffective approach. The ability to quantitatively measure how much scent-pollution of a floral-odor bumblebees can tolerate would represent a paradigm shift in odor-pollution studies. Current statistical methods for analyzing complex odors have poor predictive power because statistically-derived odor-spaces are rewritten when new odors are added. This study presents an alternative method of analyzing complex odor blends based on the encoding properties of insect olfactory systems. This "Compounds Without Borders" (CWB) method vectorizes odors in a multidimensional space representing relevant functional group and carbon characteristics of their component odorants. A single vector can be built for any scent, which allows the angular distance between any two odors to be calculated–including a learned odor and its polluted counterpart. Data presented here indicate that CWB-angles are capable of both describing and predicting bumblebee odor-discrimination behavior: odor pairs with angular distances in the 20–29˚ range appear to be generalized, while odor pairs over 30 degrees are differentiated. The neurophysiological properties underlying CWB-vectorization of odors are not unique to bumblebees; CWB-angle analysis of a small sample of published odor-data supports the idea that this method may have broader applications.

## Author summary

Recent work has indicated that anthropogenic pollution of floral-scent may have negative impacts on bumblebee foraging behavior. We need quantitative tools to both measure how much pollution of a learned floral-odor bumblebees can tolerate and identify which scent-pollutants are problematic. This study used encoding characteristics of insect olfactory systems to develop a new paradigm for quantifying complex odors. This 'Compounds Without Borders' method builds multidimensional vectors of scents based on physiologically relevant

**Funding:** The author(s) received no specific funding for this work.

**Competing interests:** The authors have declared that no competing interests exist.

physical characteristics of component odorant-compounds. The angular distance between CWB-vectors then provides a single quantitative variable describing how similar (or dissimilar) two complex odors are. This angular representation of odor similarity is predictive of bumblebees' behavior in an associative odor learning task.

## Introduction

### The ability of bumblebees to utilize sensory cues, such as floral scent, when locating resources is compromised by odor contamination

Bumblebees are prolific pollinators in both natural and agricultural ecosystems [1,2]; which makes the decades long decline in bumblebee populations [3] particularly alarming. Given that the foraging success of workers can be directly linked to reproductive output of a colony [4], understanding both general mechanisms of foraging and how anthropogenic environments impact foraging is directly relevant to conservation efforts. Flowers provide multiple sensory advertisements to pollinators; such as shape, color, and scent[5–7]. Recent computational work indicates that odor is consistently available to searching bumblebees[8] and lab-based experiments indicate that bumblebees are capable of using odor information alone to locate floral resources [9,10]. Floral-scent is likely an important sensory cue for bumblebee foragers; unfortunately anthropogenic activity has modified their olfactory landscape in urban, suburban and agricultural ecosystems [11–15]. Air pollution can have subtractive effects on scent blends by reacting with floral odorants, resulting in a modified odor structure. This reactive reduction of odorants has been shown to reduce the ability of honeybees to recognize a learned odor [12,13]. Scents can also be contaminated by the addition of novel odors, resulting in a modified odor-blend. For example, the addition of some agrochemical odors to a learned scent have been shown to modify bumblebee foraging behavior [10]. Given the vast quantity of potential odor pollutants in human habitats, it is unrealistic to approach this problem through individual behavioral tests. However, the ability of foraging bumblebees to successfully recognize then locate scattered resources may be crucial to species' survival[4]. Therefore, an effective method to predict the likelihood that a given odor pollutant will disrupt foraging behavior is needed.

### Statistical methods of measuring shifts in odor composition are descriptive rather than predictive

Current methods of describing the relative similarity (or dissimilarity) of complex odor blends rely on statistical analyses such as principal components analysis (PCA) or nondimensional metric scaling (NMDS) [16–21]. While these methods have excellent descriptive power, the dimensions of the odor spaces they create are based upon the identity of the odors' component molecules, thus all calculated dimensions are dependent upon the included data–in other words there is no quantitative independent-axis. This limits the *predictive* power of such statistical analyses, as the inclusion of a new odor/s will result in a different statistical description of the odor space, potentially changing relationships within the original data set (S1 Table).

### Is 'Compounds Without Borders', a mathematical odor-space based on the sensory energy of a scent, capable of describing and predicting odor discrimination behavior in bumblebees?

The principle goal of this study was to develop an odor-space whose quantitative-axes are independent of analyzed-scents that could be used for both descriptive and predictive analysis of

bumblebee behavior. This study proposes an odor-space whose dimensions are derived from physiologically-relevant attributes of molecules based on the current understanding of how odorants are encoded by insect-olfactory systems [16,18,22–27]. While there is little experimental evidence on olfactory processing from bumblebees [9,28], given the homology across insect olfactory systems it is reasonable to hypothesize that bumblebees have commonalities with other species [23,29]. Studies on olfactory receptor neurons (ORNs) indicate that many ORNs respond to multiple molecules sharing functional group characteristics [22,26]. Seminal work on olfactory processing in the honeybee antennal lobe demonstrated that carbon chain length and functional group are reliably encoded [18]. Their findings were supported by later work on responses to complex floral odors in hawkmoths showing that scents whose components have the same functional group elicited similar responses from the antennal lobe [16]. Moreover, recent work on the antennal lobe tracts (ALT) leaving the antennal lobe in honeybees show that the mALT carries information about functional group, while the lALT appears to encode carbon chain length [27]. The odor space presented in this manuscript is constructed of dimensions that are descriptors of volatile organic compounds' potential functional groups (such as alcohol, ester, etc) and carbon characteristics (carbon chain length and number of carbons in cyclic structures) because they should have a meaningful relationship to odor-driven behaviors. Odors are represented within this space as vectors. Vector assignment starts by characterizing the functional-group and carbon characteristics of an odor's component odorants. The relative abundance of each compound is then added to each relevant dimensions power–resulting in a single compound contributing to multiple dimensions and a single dimension's power potentially coming from multiple compounds. The angular distance between the resulting vectors of two complex odors can serve as a measurement of their similarity (or lack thereof). This odor-space is thus a mathematical representation of what types of sensory energy are contained within an odor, much as a spectrogram describes the quantity and quality of color stimuli available to a visual system. We tested the efficacy of this "Compounds Without Borders" (CWB) odor-characterization method with an associative odor learning and discrimination paradigm (free-moving proboscis extension reflex, FMPER[30]). Once the viability of the CWB-method was established, we applied it predictively–successfully hypothesizing bumblebee response to a novel set of odor-discrimination tasks. This represents a new analytical tool for both computational characterization of complex, ecologically-relevant odor blends and prediction of which types of odor pollution are more likely to disrupt odor recognition.

## Results

### Using 'Compounds Without Borders' to calculate angular distances between odor-pairs provides the ability to measure similarity with a single quantitative variable

Characterization of odor-blends entailed identifying component odorants, calculating their normalized peak areas, and determining their dimensional characteristics based upon their respective carbon chain length (CCL), cyclic carbon count (CCC), and functional groups (FG) (Fig 1, S1 Movie). The vector for each odor-blend was calculated by summing the area for all component odorants within each dimension. This 'Compounds Without Borders' (CWB) method of vector construction allows the calculation of angular distances between any two odors regardless of their underlying complexity, quantifying their relative similarity (or lack thereof) with a single variable. The CWB-angles between the three primary odors used to explore the viability of this method are shown in Fig 1C: Lily of the valley (LoV) is closer in structure to honeysuckle (HS) than juniper berry (JB), but JB is closer to LoV than HS (Table 1).

**(A)**

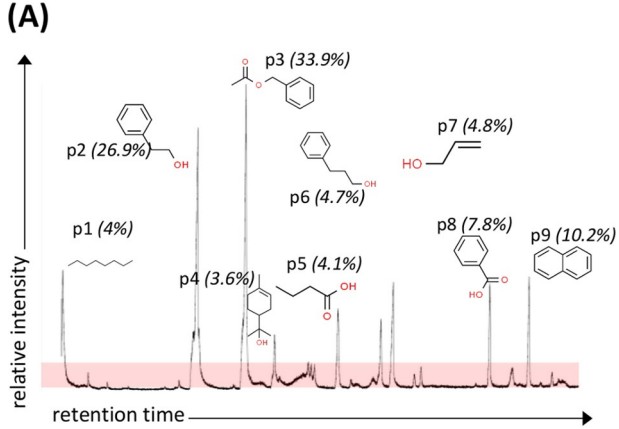

**(B)**

## Nonzero Dimensions

*Functional Group Dimensions*
Alcohol = p2+p4+p6+p7 = 40
Alkane = p1 = 4
Alkene = p7 = 4.8
Aromatic = p2+p3+p6+p8+p9 = 83.6
Bicyclic = p9 = 10.2
Carboxylic acid = p5+p8 = 11.9
Cyclic = p4 = 3.6
Cyclic Alkene = p4 = 3.6
Ester = p3 = 33.9
Methyl = p4+p5 = 37.5
*Carbon Characteristic Dimensions*
CCL1 = p4+p9 = 13.8
CCL2 = p3+p8 = 41.8
CCL3 = p2+p7 = 31.7
CCL4 = p5+p6 = 8.8
CCL8 = 4
CCC6 = p2+p3+p4+p6+p8 = 81
CCC10 = p9 = 10.2

**(C)**

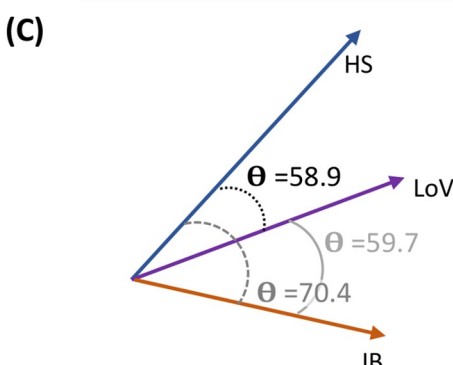

**Fig 1. CWB-vectorization of honeysuckle odor.** (A) The GC trace for honeysuckle essential oil, with peaks above 10x noise identified. The nine peaks were: p1 = octane, p2 = phenythyl alcohol, p3 = acetic acid phenylmethyl ester, p4 = α-terpineol, p5 = butanoic acid, p6 = benzenepropanol, p7 = 2-propen-1-ol, p8 = benzoic acid, p9 = napthalene. The structure of and relative area under the curve for all identified peaks is noted. (B) Each peak was characterized based on its functional groups, carbon chain length (CCL) and number of carbons in cyclic structures (CCC). A dimension receives a non-zero value if one or more odorants within an odor blend possess that characteristic. The relative peak area for all peaks with a particular characteristic are summed to determine the total power for that dimension. This method does not treat individual odorant compounds as discrete entities; rather it acknowledges that multiple different compounds may have the ability to bind to a single type of odorant receptor and that a single compound may bind multiple types of odorant receptor [22,26]. Thus by removing the boundary around individual molecules, this "Compounds Without Borders" method may more accurately model stimulation of the olfactory system. (C) The angular distance between any two CWB-vectors can be calculated, allowing the similarity or dissimilarity of two odors to be represented with a single quantitative variable.

## FMPER is an effective method for measuring odor learning

We modified the free-moving PER method presented by Muth et al. [30] to allow for odor stimulation (Fig 2A). In brief bumblebees are trained to associate an odor with a sugar reward presented on a scented-plastic strip over the course of four conditioning trails, after which they are given a choice of two scented-, unrewarding-strips: one with the associative odor (AO) and one with a contrasting odor (CO). Bumblebees participating in FMPER had four potential outcomes: 1. they could fail to complete four conditioning trials and be dropped from analysis; 2. they could choose correctly, extending their proboscis in response to the AO- strip; 3. they could choose incorrectly, extending their proboscis in response to the CO- strip; or 4. they could not choose. Bumblebees were classified as "no-choice" if they successfully completed four conditioning trials but did not choose after interacting with the test strips three times. 446 individual bees from 10 *Bombus impatiens* colonies were tested in these FMPER experiments. Of these 446, a total of 89 individuals (20%) were excluded from analysis, leaving 357 for analysis: 27 (6%) did not complete all four training trails; 17 (4%) were removed because of experimenter or equipment error; and 45 (10%) were from individual experiments resulting entirely in 'no-choice' (unfiltered data available for download in supplemental materials).

Bumblebees demonstrated an ability to discriminate between an associative odor (AO) and unscented mineral oil (Fig 2C): those associated to LoV yielded 66.7% correct (p = 0.0006; exact test against random distribution) and association to juniper berry (JB) showed 71.4% (p = 0.002). These data were combined with results from tests of 5 additional types of associative odors (see Methods, S2 Table) to calculate an expected response distribution for the FMPER assay when bees were given a high-contrast odor discrimination task (a perceivable AO versus unscented-mineral oil). Responses to lower-contrast odor discrimination tasks were tested against the expected distribution of 61.6% correct, 8.5% incorrect, and 29.9% no choice. Tests that returned p-values less than 0.15 are subjected to binomial post-hoc comparisons of the correct, incorrect and no-choice respnses. Following the recommendations of Amrhein et al., we are reporting exact p-values and not classifying data into binary categories of 'significant' versus 'insignificant'[31]. Data on odor choices yielding small p-values for statistical comparisons with the expected distribution *and* post-hoc percent correct are interpreted as indicating that the two tested odors are less easily discriminated, and likely being treated similarly–a phenomenon referred to as generalization [32].

## FMPER responses indicate that CWB-angles can identify a threshold for discrimination

Bumblebees associated to Lily of the Valley (LoV) were tested against: LoV, honeysuckle (HS), juniper berry (JB) and a range of corresponding blends (creating contrasting odors that were

**Table 1. Stimulus characteristics and response details for all discrimination tasks.** The 'random' distribution assumes an equal probability of correct, incorrect, and no-choice responses, while 'expected' represents the theoretical response. C = correct, I = incorrect, and NC = no choice.

| Associative Odor | Comparison Odor | Angle | p (vs random) | p (vs expected) | % C/I/NC | n | PostHoc Test: % Correct | PostHoc Test: % Incorrect | PostHoc Test: % no choice |
|---|---|---|---|---|---|---|---|---|---|
| LoV | MO | n/a | 0.0006 | 0.22 | 66.7/15.2/18.2 | 33 | n/a | n/a | n/a |
| LoV | LoV | 0 | n/a | 0.01 | 27.8/22.2/50 | 15 | 0.01 | 0.06 | 0.07 |
| LoV | 3 Lov: 1 JB | 15.6 | n/a | 0.03 | 47.6/0/52.4 | 21 | 0.26 | 0.25 | 0.03 |
| LoV | 1 LoV: 1 HS | 28.8 | n/a | 0.04 | 33.3/6.7/60 | 15 | 0.03 | 1 | 0.02 |
| LoV | 1 LoV: 1 JB | 32.2 | n/a | 0.51 | 53.3/20/26.7 | 15 | n/a | n/a | n/a |
| LoV | HS | 58.9 | n/a | 0.49 | 60/20/20 | 15 | n/a | n/a | n/a |
| LoV | JB | 59.7 | n/a | 0.18 | 61.9/0/38.1 | 21 | 1 | 0.25 | 0.47 |
| JB | MO | n/a | 0.002 | 0.28 | 71.4/14.3/14.3 | 21 | n/a | n/a | n/a |
| JB | JB | 0 | n/a | 0.003 | 22.2/22.2/55.6 | 18 | 0.001 | 0.06 | 0.03 |
| JB | 3 JB: 1 LoV | 12.3 | n/a | 0.04 | 56.7/0/43.3 | 30 | 0.58 | 0.18 | 0.11 |
| JB | JB: 1 LoV | 27.5 | n/a | 0.09 | 33.3/20/46.7 | 15 | 0.03 | 0.13 | 0.17 |
| JB | 1 JB: 1 HS | 31.5 | n/a | 0.79 | 53.3/13.3/33.3 | 15 | n/a | n/a | n/a |
| JB | LoV | 59.7 | n/a | 0.33 | 70.6/0/29.4 | 17 | n/a | n/a | n/a |
| JB | HS | 70.4 | n/a | 0.37 | 64.7/0/35.3 | 17 | n/a | n/a | n/a |
| Crd | MO | n/a | 0.033 | 0.30 | 50/13.3/36.7 | 30 | n/a | n/a | n/a |
| Crd | 1 Crd: 1 Brg | 21.2 | n/a | 0.01 | 38.9/0/61.1 | 18 | 0.05 | 0.40 | 0.01 |
| Crd | Brg | 45.0 | n/a | 0.23 | 48.6/8.1/43.2 | 37 | n/a | n/a | n/a |

**Key to abbreviations:** MO = unscented mineral oil; LoV = lily of the valley; JB = juniper berry; HS = honeysuckle; Crd = cardamom; Brg = bergamot

polluted versions of the associative odor) that resulted in an angular range of 0–59.7˚. Blends are abbreviated with their relative ratios and identities, such that a blend with equal volumes of honeysuckle and juniperberry would be notated as 1HS:1JB. Bumblebees associated to JB were likewise tested against JB, HS, LoV and a range of corresponding blends that resulted in an angular range of 0–70.4˚. In both AO conditions bumblebee response distributions to the 0˚ task returned a very low p-value when compared to the expected distribution (LoV vs LoV, p = 0.01, n = 15; JB vs JB, p = 0.003, n = 18). Interestingly, both these tests showed a larger than expected percent no-choice: 50% for LoV-associated bumblebees and 55.6% for JB-associated (Fig 3). In both AO conditions, responses to discrimination tasks with an angular distance larger than 30˚ had distributions that were similar to the expected response, with p values ranging from 0.18–0.51 (Fig 3, Table 1), with high percent correct (>50%); indicating that the AO and CO were easily discriminated. Reponses to angles below the 30˚ threshold were more complicated. Bumblebees given AO vs CO tests with distances in the 20–29˚ range showed response distributions that were similar to the 0˚ tests: low percentages correct (< 40%), higher percentages of no-response (46.7–61.1%) and smaller p-values (Fig 3, Table 1) indicating that the CO was likely generalized to the AO. For both the LoV vs 1LoV:1HS (θ = 28.8, p = 0.04) and the JB vs. 1JB:1LoV (θ = 27.5, p = 0.09) post-hoc tests support the observed decrease in percent correct (binomial test, p = 0.03 and p = 0.03 respectively). However, in contrast with results from 0˚ choices, these data did have higher percent correct than incorrect. Responses to

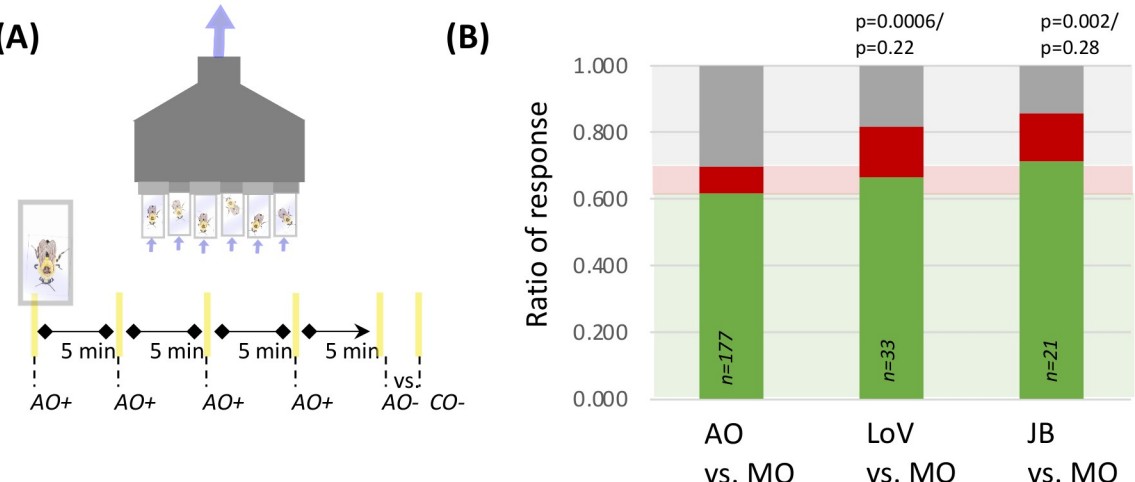

**Fig 2. FMPER is an effective associative odor-learning paradigm.** (A) The ventilating array held 6 bees in screen-backed vials, bringing air in through two lid-holes and out the back. A sucrose reward was delivered on yellow associative odor (AO)-scented plastic strips during the four conditioning trials. The testing trial presented bees with two unrewarding strips (AO vs contrasting odor (CO)) and recorded bumblebee responses as correct, incorrect, or no-choice. (B) The percentage of correct (green), incorrect (red), and no-choice (grey) responses from 7 different associative odor (AO) versus unscented mineral oil tests, utilizing a total of 177 bees, were used as the theoretical response distribution for high-contrast odor choice in this FMPER paradigm. This distribution is labeled "AO vs MO". The results of AO versus mineral oil tests for lily of the valley (LoV) and juniper berry (JB) are also shown, with the background green, red and grey boxes representing the expected response distribution. The p-values for comparisons to a random distribution (top) and the expected distribution (bottom) are noted. The sample sizes are included on each bar.

angular separations of 12–16° show an increase in percent correct and a complete absence of incorrect choices. For the JB vs 3 JB:1LoV test ($\theta$ = 12.3°, p = 0.04), with 56.7% correct, and the LoV vs 3LoV:1JB test ($\theta$ = 15.6°, p = 0.03), with 47.6% correct, post-hoc tests do not indicate a decrease in the percent correct responses from expected (p = 0.58 and p = 0.26 respectively). In these cases the statistical difference from expected may be driven by the percentage of no choice responses (43.3% for AO = JB and 52.4% for AO = LoV, p = 0.11 and p = 0.03 respectively).

## The CWB-method is capable of predicting bumblebee-responses in a FMPER task

The principle goal in developing the CWB-method was to facilitate characterization of odors in a manner that provided some predictive power for *B. impatiens* responses. Thus when preliminary data analysis indicated a putative threshold of generalization (approximately 30°), the CWB method was used to select two additional odor discrimination tasks: one with a sub-threshold angular distance between the AO and CO, and one over threshold. The subthreshold task should yield a response distribution that differs from expected, while the suprathreshold task should be similar to the expected distribution. Using cardamom essential oil as the AO, the sub-threshold CO was 1 cardamom: 1 bergamot ($\theta$ = 21.2°), and the suprathreshold CO was bergamot essential oil ($\theta$ = 45°). Indeed, the distribution for the 45° task appears to match the expected easy odor choice distribution (p = 0.23), while the distribution for the 24.8° task differs (p = 0.01) (Fig 4)–thus the tested responses matched the apriori predictions.

## Scent contamination reduces odor discrimination ability

The threshold for generalization cuts right through the cluster of COs that were constructed by blending a contaminating scent with the AO (Fig 5). In these cases, bumblebees were clearly able to discriminate between the uncontaminated AO and its polluted CO, indicating that the

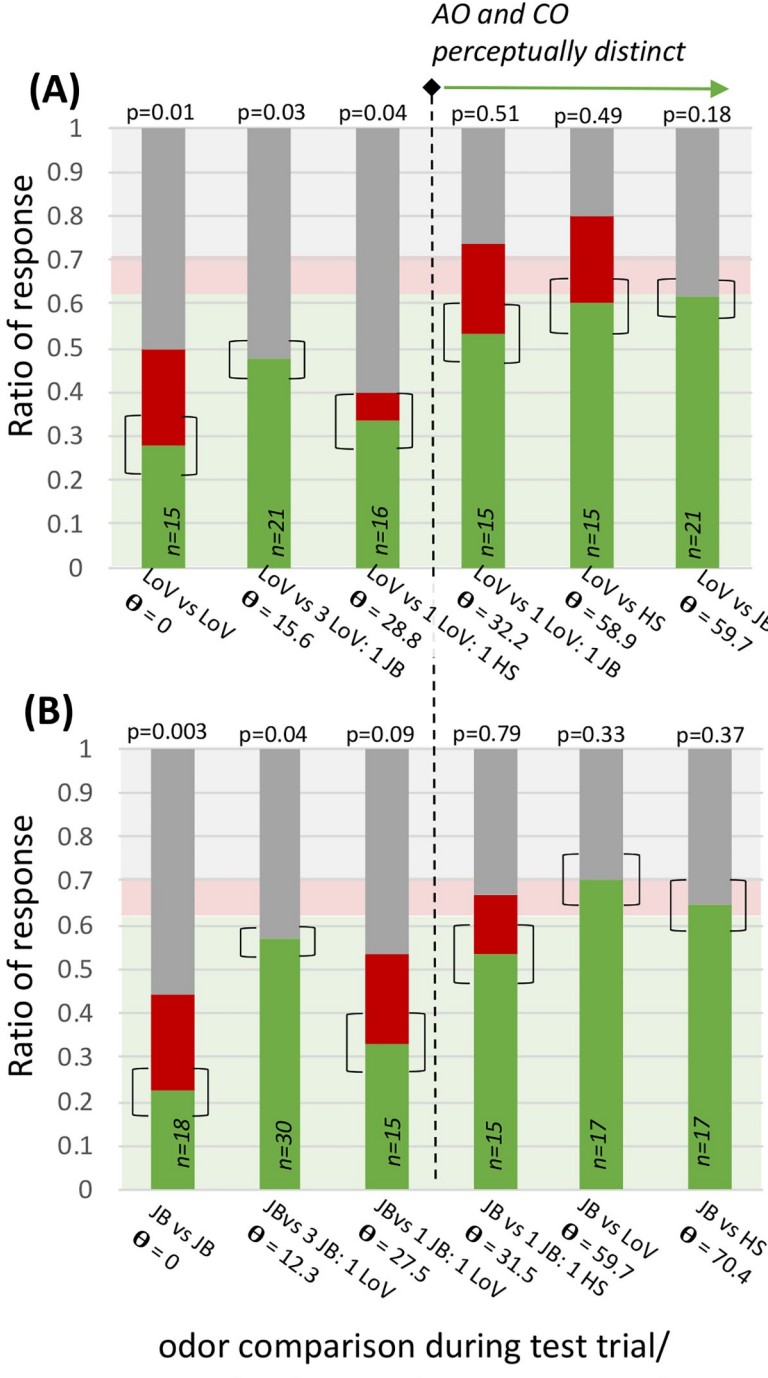

**Fig 3. Comparison of CWB-angles to bumblebee responses indicates a discrimination threshold of 30˚.** The response distributions for bumblebees associated to lily of the valley (A) and juniper berry (B) to discrimination tasks against contrasting odor (CO) stimuli with increasing angular distances to the associative odor (AO). Green portions of bars represent correct responses, red represent incorrect, and grey represent no choice. The transparent green, red and grey boxes in the background represent the expected response distribution. The p-values for the statistical comparisons with this distribution and sample sizes are noted for each task. All odors above 30˚ appear to be easily discriminated, as indicated with the vertical dashed line. The brackets around the correct response ratios represent the difference one bee makes (i.e. if one more or one less bee chose correctly).

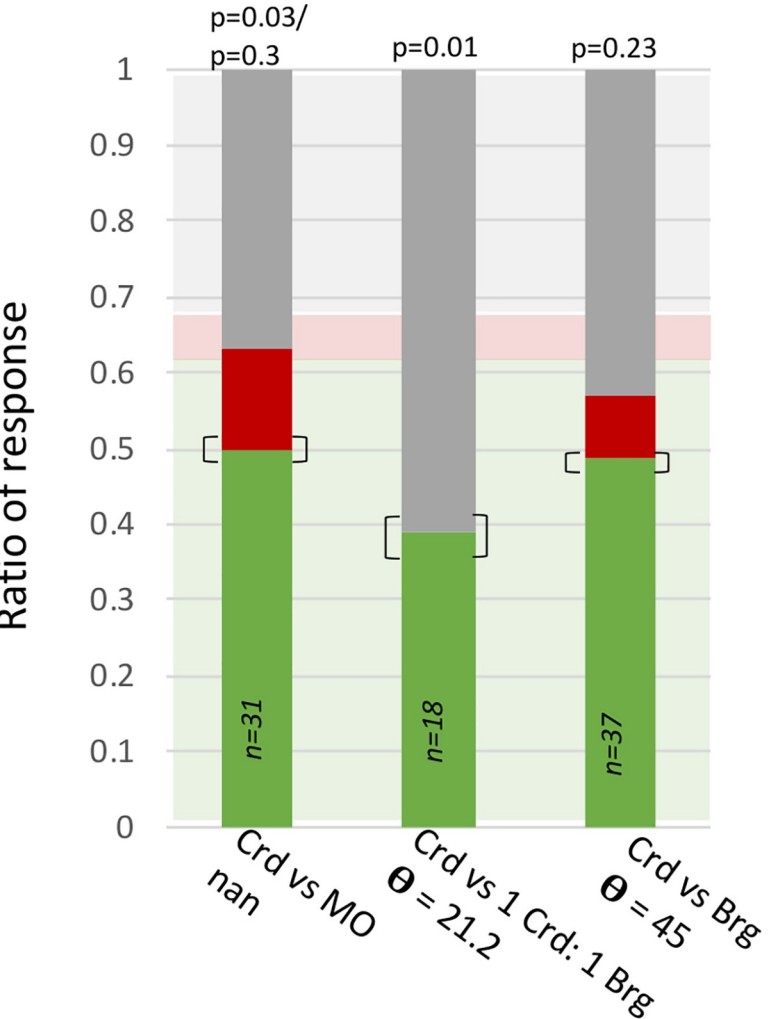

**Fig 4. The response distributions for odor discrimination tasks testing the putative 30˚ discrimination threshold.** Bumblebees were associated to cardamom (Crd) essential oil and tested against: mineral oil (confirming bumblebees are capable of learning Crd), a subthreshold CO (θ = 21.2˚, Crd vs 1 Crd: 1 bergamot (Brg)), and a suprathreshold CO ((θ = 45˚, Crd vs Brg). Green portions of bars represent correct responses, red represent incorrect, and grey represent no choice. The transparent green, red and grey boxes in the background represent the expected response distribution. The brackets around the correct response ratios represent the difference a change in one bee's response would make.

polluted version of the learned odor was no longer being treated as the AO itself. However, bumblebees appear to have tolerance for scent contamination in the 20–29˚ range. Bumblebees tested with COs from this range showed a decreased percent correct and a departure from the expected response distribution, indicating the contaminated scent was difficult to discriminate from the original and likely generalized.

## Discussion

### Compounds without borders effectively describes olfactory discrimination behavior in an associative learning context

While the role of odor signals in plant-pollinator relationships is not completely understood, numerous studies indicate that odor is an important sensory modality in bumblebee foraging

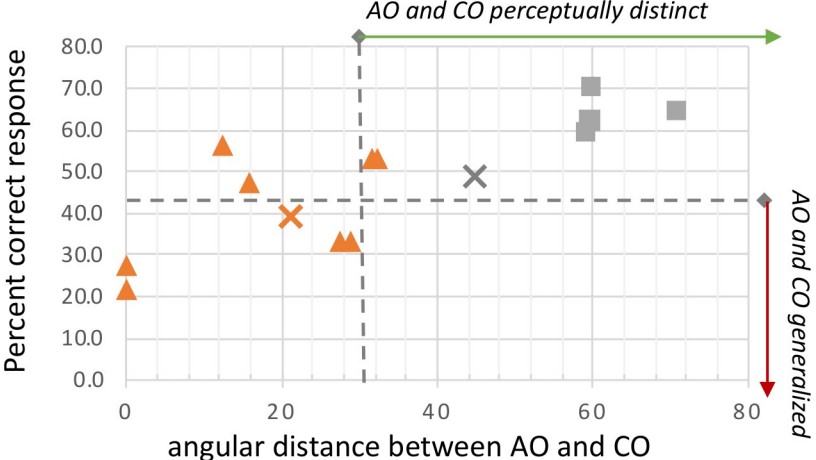

**Fig 5. All contrasting odors greater than 30˚ from the associative odor are discriminated, even if they are polluted versions of the AO.** The percent correct data from all discrimination tasks replotted against the angular distance between the AO and CO. Data are differentiated by whether the CO was a blend of the AO with a polluting odor (σ) or if the CO was a distinctly different scent (ν). The predictive data (AO = cardamom) are represented with x's: an orange **X** for AO-blended CO and a grey **X** for the distinctly different CO. Tests with % correct values below the horizontal dashed line consistently had small p-values when compared to the expected correct response percentage. The vertical dashed line at 30˚ represents the threshold above which bumblebees can easily discriminate between the AO and the CO.

[9,33–35]. Given the complex olfactory landscapes that pollinators operate within [36], discrimination thresholds could have consequences for foraging efficiency. In an environment where target flowers (i.e. those a pollinator has coevolved with) have similar scents [16], foragers with a larger discrimination threshold-angle and a generous generalization range could maximize feeding opportunities. Under the baseline conditions in this study (absolute conditioning and discrimination tests without an odor background) bumblebees showed an apparent generalization range from 20–29˚; with all tests in this range demonstrating a significantly lower percentage of correct responses than expected. Odors above a 30˚ threshold were consistently discriminated from the associative odor, regardless of whether or not the contrasting odor was a blended version of the AO. These angular ranges were effective descriptors of bumblebee behavior when bees were associated to two different complex odor blends, lily of the valley and juniper berry, *and* served as an effective predictor of behavior for bumblebees associated to cardamom. However, the discrimination threshold implied here does not likely represent a perceptual limitation. Work on ants has shown that previously generalized odor pairs can be discriminated after training with differential conditioning, implying that the behavioral-discrimination measured was contextual [32]. Indeed, the role of experience in generalization behavior has been demonstrated in multiple studies [37,38]. This behaviorally-demonstrated ability to dynamically shift discrimination thresholds in insects has been supported by work showing that learning can modulate neural activity within the antennal lobe [16,17]. Within the dataset presented here, responses to AO vs CO tasks with an 11–16˚ separation also support the idea that the 20–29˚ generalization range does not represent a perceptual limitation. Angles within this close range do not always appear to be clearly generalized. For both the 12.3˚ (AO = JB, CO = 3JB: 1LoV) and 15.6˚ (AO = LoV, 3LoV: 1JB) tests the overall response distribution returned low p values (0.03 and 0.04 respectively) when compared to the expected distribution. However, post-hoc tests indicate that the percentage of correct responses was not lower than expected (p = 0.58 and 0.26 respectively). This could imply that there is a perceptually-distinct but behaviorally-ambiguous range where responses are less predictable. There are several potential ecological and ethological reasons why a range of

ambiguity might exist. For example, recruitment pheromone activates foraging in bumblebee workers [39,40], and newly activated foragers are more likely to seek out floral odors brought into the hive by a returning worker [33]. Foraging pheromone would increase the power of eleven different CWB dimensions (S3 Table), three of them with contributions from multiple components [33]. Depending on the volume released, the angular shift induced by phero-monal-'pollution' of floral odors brought in by returning foragers is likely to be small. Newly recruited foragers would want to generalize that small angular shift to the original floral blend, otherwise they risk not recognizing that floral resource. However, some small angular shifts might be disadvantageous to generalize. Recent work on volatile emission from microbial nec-tar communities has shown differences in their odors. Interestingly honeybees show differen-tial attraction to microbial scents [19]. If the angular shift in floral odor induced by a microbial community is small, *and* if the added odor is indicating the presence of an undesirable microbe, than generalizing the microbe-polluted floral odor to the original would be disadvan-tageous. The ecological, ethological, neurophysiological, and perceptual drivers of behaviors to odor shifts in this angular range are fascinating topics for future study.

## Understanding generalization and discrimination behavior of bumblebees could help conservation efforts to reduce the impact of odor pollution

As of June 2019 Home Depot's website listed 333 products under plant care geared for disease control and fertilization. At least one prior study has shown that the scent of a consumer lawn product is capable of modifying bumblebee foraging behavior in the lab [10]—but piecewise testing of 333 products is an intractable amount of work for the academic community and agrochemical producers are unlikely to take up the cause. Agrochemical odor pollution is not confined to consumer products; commercial agrochemicals may also be problematic [10]. Using the CWB-method to calculate angular shifts due to agrochemical scent-pollution could identify products that shift odors into a 'zone of concern', namely outside the identified gener-alization zone. This could be a remarkably useful tool for predicting which products are likely to disrupt bumblebee recognition of a learned floral resource. Likewise, this may provide a strategy for encouraging bumblebees to avoid resources recently treated with insecticides. However, all real-world ecological applications will require field testing.

Unfortunately, agrochemicals are not the sole source of odor pollutants that pollinators contend with–previous work has shown that air pollutants, such as diesel exhaust and ozone, react with floral odorants[11–15]. This reduces the distance that floral odors travel, which could have impacts on signal encounter by searching foragers [8,12]. In addition, the reaction of floral odorants with air pollutants changes the blend structure–in some cases pushing a learned odor far enough that honeybees no longer respond normally [13–15]. CWB-angles may provide a useful tool for investigating what levels of air pollution are likely to disrupt learned odor responses, and which still allow generalization to known scents.

## Potential applications and limitations of the CWB-method

Due to the limited neurophysiological data from bumblebee olfactory-systems [9,28] the archi-tecture of the CWB-dimensions was based on the encoding properties of other insect species [41], [18,42], [26,43]. Therefore, the CWB-method may be applicable to insects more broadly. Analysis of a small sample of both floral scent structure and pollinator olfaction studies within the literature indicates that CWB-angles are consistently smaller between odors that clustered more closely together in the original studies' statistical analyses (Fig 6) [16,20,21]. CWB-angles by definition quantify the 'distance' between scents; across all 6 studies analyzed interspecies angles ranged from 6˚ to 88˚[6,16,20,21,32,44], indicating the distances between stimuli in this

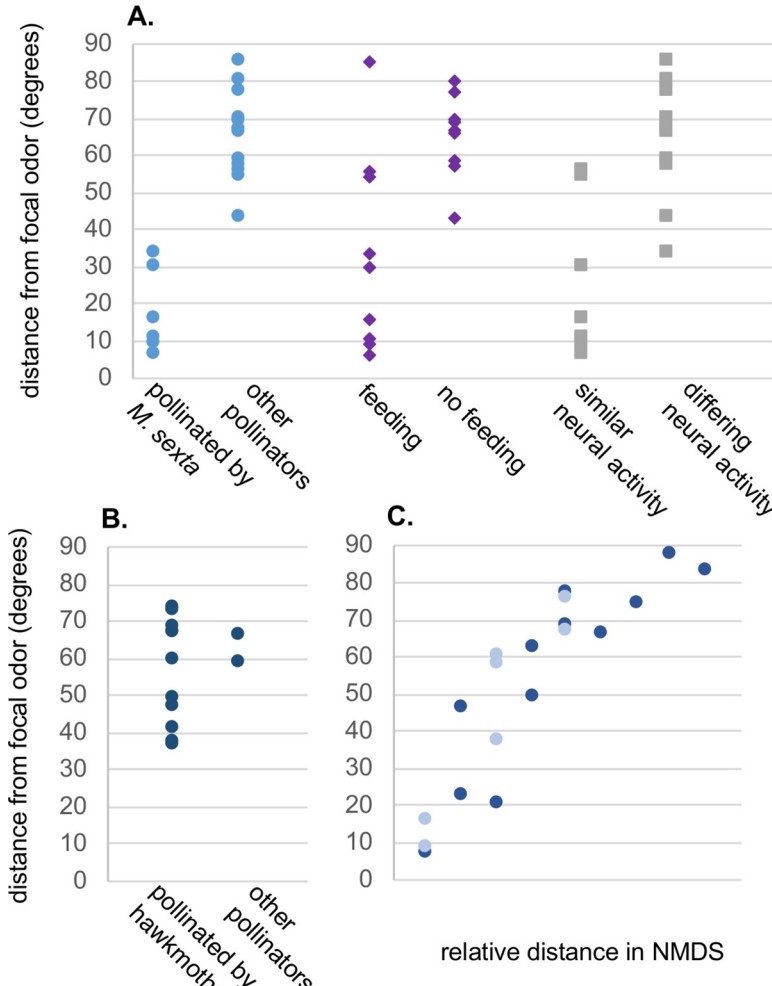

**Fig 6. Application of CWB-angle analysis to a selection of published studies on odor-structure and insect behavior.** (A) Comparison of CWB-angles to the original PCA from Riffell et al. indicates they are capable of recreating traditional approaches to complex odor-blend analysis [16]. Riffell et al. used PCA determine relationships between odor-blend structures (blue circle), as well as the relationship between odor and hawkmoths' innate behavioral (purple diamond) and neural (grey square) responses. A hawkmoth-pollinated flower, *Nicotiana suaveolans*, from their data was randomly selected as the anchoring odor, and the CWB-angles to all other floral-odors they presented were calculated and plotted to the y-axis. The left hand side of each category shows odors/ responses that were clustered in their PCA, while the right hand side shows odors/ reponses that were outside their identified clusters. The 35˚ threshold between odors similar to *N. suaveolans* and those that were not is largely descriptive of both behavioral and neural responses. (B) CWB-angles were calculated between *Mirabilis longiflora* and 13 other species from the family Nyctaginaceae from data published by Levin et al [44]. All hawkmoth pollinated species (confirmed by author observation or prior literature) are plotted on the left, while the non-hawmoth pollinated species are on the right. (C) A comparison of CWB-angles to the statistical analysis (NMDS) of two different studies that characterized the scent-composition of flowers and fungi in the context of pollination indicate that odors clustering more closely in statistical space also have smaller CWB-angles. Lighter blue circles represent data from Jürgens et al [21], while the darker blue circles represent data from Johnson and Jürgens [20].

study are in an ecologically relevant range. In some scenarios a more nuanced analysis of how scents differ might be relevant, such that a simple angle does not provide enough information. Comparisons of the CWB-vectors themselves can provide additional detail. For example when looking at data from Levin et al 2001, which characterized the headspace of 20 different species in the family Nyctaginaceae, there is not a strong difference in angular distance from *Mirabilis longiflora* based on a shared pollinator (Fig 6) [44]. However, a heatmap analysis of the CWB-

vectors shows that the one bee-pollinated species analyzed has more power in the alkane and cyclic dimensions than any of the other odors (S1 Fig). Of the six studies analyzed, three collected data on the relationship between olfactory stimuli and insect behavior (in hawkmoths [16], bumblebees [6], and ants[32]). Riffell et al. assembled a comprehensive dataset of odors from a phylogenetically diverse set of flowers in a seminal study on the coevolutionary and neurophysiological relationships between hawkmoths and hawkmoth-pollinated flowers. CWB-angle analysis of the sampled odors identified a 35˚ threshold for closely clustered odors in the original PCA. Moreover, with few exceptions, CWB-angles also correlate with their PCA analyses of neurophysiological and behavioral data (Fig 6) [16]. The second study analyzed examined the effect of modifying the scent structure of differing snapdragon flower-morphs on bumblebee visitation rates. White flowers were spiked with methyl benzoate (produced by yellow flowers) and yellow flowers were spiked with methyl cinnamate (produced by white flowers). The scent distance between unmodified yellow and white snapdragon morphs was 36.3˚, and the authors measured a difference in visitation rates between these two morphs. Interestingly, scent modification of the morphs, which induced shifts of 4–5˚, did not change visitation rates[6]. These field measurements align with the results presented in this manuscript, with scents greater than 30˚ being treated as different, while those under 30˚ have a high likelihood of being generalized. Based on this cursory literature analysis the CWB method may be effective at describing the relationship between scent-structure and pollinator-behavior more broadly. However, this methods's odor-dimensions broadly characterize molecules and its demonstrated efficacy is potentially due to its ability to describe the emergent properties of varying binding probabilities at the level of olfactory receptors, lateral processing between olfactory sensory neurons and inter-glomerular processing at the antennal lobe [23,29]. These emergent properties will only come into play with complex, multi-component olfactory stimuli. Odor-driven behaviors, such as pheromone tracking, that are based on simpler-stimuli with smaller numbers of odorants may be less effectively characterized by CWB-angles. Indeed, application of CWB-angles to work by Perez et al on generalization and overshadowing in ants indicates just that. As part of their study, Perez et al trained ants to nonanal in an associative learning task, then tested for generalization with three different binary blends (1 nonanal: 1 hexanal, heptanal, or octanal). The CWB-angle between nonanal and all three blends was identical (24.1˚), but the ants only generalized to two of the blends (heptanal and octanal) [32]. Thus CWB-angles appear more effective for complex odor stimuli than simple ones. While the work presented here provides proof of concept for 'compound without borders', its inter-species and ecological applications will require additional experimental validation.

## Methods

### Odor stimuli

We selected three essential oils (*New Directions Aromatics)* to serve as odor-blends for this study: lily of the valley, honeysuckle and juniper berry. Lily of the valley (LoV) has been successfully used in odor learning experiments utilizing the proboscis extension reflex (PER) [45], and the two additional odors were selected to provide a range of structural overlap with LoV. For further discussion of odor stimuli selection, see supplemental materials. Blending these three in varying ratios allowed construction of odor stimuli with varying ranges of odorant composition and distance (Table 1). These selected essential oils, as well as those from the predictive FMPER experiments, were sampled using Solid-Phase Microextraction (SPME) fibers and their composition analyzed with GCMS. For full details please see supplementary methods.

## CWB-vectorization of odor-blends and calculation of angular distances

Characterization of odor blends started with identification of component odorants and calculation of their normalized peak areas. The dimensional signature of each odor-blend was then determined by the molecular structure of its component odorants based upon their respective carbon chain length (CCL), cyclic carbon count (CCC), and functional group (FG) characteristics (Fig 1). From a dimensional perspective, the power for each dimension was calculated as the summed area of all peaks from molecules with that attribute (CCL, CCC, or FG). If no molecules within a given odor blend have that attribute, that dimension has a power of zero. From a peak/ molecule perspective, each odorant's normalized area will be assigned to multiple dimensions; with a minimum of two (CCL and at least one FG) and no maximum (Fig 1).

The CWB method represents odors as vectors in a 66-dimensional space. This vector representation allows us to calculate angular differences between odor blends, where given two odor vectors a and b the angle between them can be calculated as:

$$\theta = cos^{-1}(a \cdot b/|a| \times |b|)$$

The calculated vectors and the odor blend classifications are available in the supplemental materials (S1 Dataset) and the R code for calculating vectors and angles are available in the S1 Appendix. This method is similar to that of Snitz et al, which used vectors from a set of 21 chemical descriptors that were optimized from an original set of 1483 descriptors from proprietary software (Dragon) to calculate angles between odor blends; which were then correlated with human perceptual descriptions[46]. The principle differences between the Snitz et al study and this study is that the CWB-method of dimensionalization is based upon known properties of olfactory processing (both at receptor and higher-order levels) and does not require statistical optimization.

## Associative testing of odor discrimination with FMPER

Testing the efficacy of the CWB-method of odor representation required a reliable method of assessing odor learning and discrimination. We modified the free-moving PER method presented by Muth et al. [30] to allow for odor stimulation (Fig 2A). Healthy-, active-individual *Bombus impatiens* (from Kopert Biological) were selected from lab colonies, placed in screen backed vials, acclimated for two hours, and placed into the odor stimulation apparatus. The ventilating testing array drew air in through two small holes in the lid and out the back, with flow rates ranging from 0.1–0.3 m/s (VWR-21800-024 hot wire anemometer). During conditioning bees were offered a single drop of sucrose on a yellow strips cut from plastic folders, which had absorbent adhesive bandage tape (Cover Roll) placed on the back to hold associative odor stimuli (1 μL of essential oil). The plastic prevented the odor-solution from diffusing into the sugar solution on the tip of the strip, therefore the primary sensory encounter with odorants was through the olfactory rather than the gustatory system. Four conditioning trials were followed by an unrewarded test trial with five minute inter-trial intervals (Fig 2A). Conditioning trials were not started until a bee successfully consumed sucrose. Test trials presented conditioned bees with two unrewarding strips- one scented with the associative odor (AO) and the other with the contrasting odor (CO). The complete protocol for these experiments can be found in the S2 Appendix. The contrasting odors are detailed in Table 1. Bumblebees participating in FMPER had four potential outcomes: 1. They could fail to complete four conditioning trials and be dropped from analysis; 2. They could choose correctly, extending their proboscis on the AO strip or while antennating the AO strip; 3. They could choose incorrectly, extending their proboscis on the CO strip or while antennating the CO strip; or 4. They could not choose. Bumblebees were classified as "no-choice" if they successfully completed four

conditioning trials but when presented with two unrewarded strips during a test trial approached and investigated strips three times without choosing. On the rare occasion that all bees tested on a given day returned 'no-choice', those data were excluded from the overall data set (45/446 bees). All FMPER data are available in the supplemental materials (S2 Dataset).

## Statistical analysis of FMPER results

Statistical analyses of FMPER results asked two questions: 1) does FMPER provide a reasonable measure of associative olfactory learning; and 2) are two odors difficult to discriminate from each other?

To determine if FMPER tests associative olfactory learning, bumblebees' responses to a discrimination task of the AO from unscented mineral oil (MO) were tested against random chance with an exact test of goodness of fit [47].

Given that FMPER did indeed provide a reasonable measure of odor-learning (Fig 3), the response-distribution of bumblebees tested with an AO vs MO should represent a high-contrast, simple odor discrimination task. We used data from the three AO vs MO experiments in this study (AO = lily of the valley, juniper berry, or cardamom), as well as four additional AO vs MO experiments from a separate methods study (AO = commercial peppermint oil, lab distilled mint oil, or mint leaves; Edwards et al. *in prep*) to calculate the mean correct, incorrect and no-choice responses made by bumblebees in this simple task (S2 Table). This was used as the theoretical response distribution of how bumblebees respond to a high-contrast odor discrimination task in the FMPER assay (Fig 2B). Further details are available in the supplemental materials. In order to answer the question, 'were the two tested odors (the AO and the CO) difficult to discriminate?' we tested response distributions for each test against the theoretical distribution (61.6% correct, 8.5% incorrect, and 29.9% no choice) with an exact test of goodness-of-fit using a log likelihood ratio method for calculating p values. Therefore the null hypothesis (accepted if p values are large) would be that the two odors are in fact easily discriminated. Rejection of the null hypothesis would indicate that the odors are not easily discriminated, or are generalized. Readers wanting to assess traditional 'significance' of p-values of these analyses may consider using an alpha level of 0.05. Following the recommendations of Amrhein et al., we are reporting exact p-values and not classifying data into binary categories of 'significant' versus 'insignificant'[31].

## Supporting information

**S1 Methods. Supplementary methods.**
(DOCX)

**S1 Table. A comparison of the angular distance between juniper berry (JB) and lily of the valley (LoV) using PCA dimensions and CWB dimensions.** For each method the angle was calculated with two different datasets included in analysis. The 'subset' was the original set of three odorants tested in this manuscript: honeysuckle (HS), juniper berry (JB) and lily of the valley (LoV). The complete dataset ('All') also included bergamot and cardamom. Because PCA calculates statistical dimensions based upon the input data, the angle between JB and LoV changes with the addition of new odors. CWB angles are based on constant vectors and do not change with additional odors in the analysis.
(DOCX)

**S2 Table. Data used to calculate the theoretical response distribution for a high contrast odor choice in FMPER experiments.**
(DOCX)

**S3 Table. Dimensional attributes of synthetic foraging pheromone constituents.**
(DOCX)

**S1 Movie. Animation of Fig 1.** This movie provides a video walkthrough of CWB-vector construction for honeysuckle essential oil; the same data from Fig 1 in the main manuscript.
(MOV)

**S1 Fig. Heatmap analysis of CWB-vectors for the twenty Nyctaginaceae species.** CWB-vector analysis of the twenty different Nyctaginaceae species characterized by Levin et al 2001[4] is shown below. The columns are plant species, while the rows are CWB-dimensions. The heatmap and linkages were created with the heatmap() function in R. The one bumblebee-pollinated species is denoted with a bee. The species key for abbreviations is: Aac = *Acleisanthes acutifolia*, Acr = *A. crassifolia*, Alo = *A. longiflora*, Aob = *A. obtuse*, Awr = *A. wrightii*, Mal = *Mirabilis alipes*, Mbi = *M. bigelovii*, Mgr = *M. greenei*, Mja = *M. jalapa*, Mlo = *M. longiflora*, Mmf = *M. macfarlanei*, Mmu = *M. multiflora*, Mpu = *M. pudica*, Mtr = *M. trifloral*, San = *Selinocarpus angustifolius*, Sch = *S. chenopodiodes*, Sla = *S. lanceolatus*, Spa = *S. parvifolius*, Spu = *S. purpusianus*, Sun = *S. undulates*
(TIF)

**S1 Appendix. CWB code.** This appendix contains text with all R code necessary to calculate CWB-vectors and angles.
(DOCX)

**S2 Appendix. FMPER protocol.** This appendix contains the FMPER protocol for data collected in these experiments.
(DOCX)

**S1 Data. Odor data.** This dataset contains an .xlsx file where each sheet can be copied to an individual .csv file to recreate the presented CWB-analysis. It has the CWB-vectors for Brg, Crd, Hs, JB, and Lov, the individual files of each odor's structure, and the main compound database.
(XLSX)

**S2 Data. FMPER data.** This dataset contains all data collected for the FMPER experiments on the first sheet (with color codes for filtering conditions described in the methods section). The second sheet has the filtered data.
(XLSX)

## Acknowledgments

Thank you to Nick Roma for his GC-MS work, and to Alexandra Domardsky and Sara Kass for their work on establishing FMPER protocols in the lab. Abigail Edwards, Katie Esbenshade, Vanessa Pham, Jessica Sommer, Baadal Vacchani, Katie Chen, Natalie David, Rachel Koerwer, Vijay Rao, and Morgan Tietz assisted in FMPER data collection. Many thanks to Dr. Allison Davidson and Dr. James Russell for statistical consultation. This work would not have been possible without consultations with Dr. Tim Henry and Dr. Christine Ingersoll's assistance on GC-MS protocols and operations.

## Author Contributions

**Conceptualization:** Jordanna D. H. Sprayberry.

**Data curation:** Jordanna D. H. Sprayberry.

**Formal analysis:** Jordanna D. H. Sprayberry.

**Methodology:** Jordanna D. H. Sprayberry.

**Project administration:** Jordanna D. H. Sprayberry.

**Resources:** Jordanna D. H. Sprayberry.

**Supervision:** Jordanna D. H. Sprayberry.

**Visualization:** Jordanna D. H. Sprayberry.

**Writing – original draft:** Jordanna D. H. Sprayberry.

**Writing – review & editing:** Jordanna D. H. Sprayberry.

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
