## [Decision Letter · Decision Letter 0]

22 Aug 2019

Dear Dr Sprayberry,

Thank you very much for submitting your manuscript 'Compounds without borders: a novel paradigm for quantifying complex odors and responses to scent-pollution in bumblebees' for review by PLOS Computational Biology. Your manuscript has been fully evaluated by the PLOS Computational Biology editorial team and in this case also by independent peer reviewers. The reviewers appreciated the attention to an important problem, but raised some substantial concerns about the manuscript as it currently stands. While your manuscript cannot be accepted in its present form, we are willing to consider a revised version in which the issues raised by the reviewers have been adequately addressed. We cannot, of course, promise publication at that time.

Your revisions should thouroughly address all of the specific points made by each reviewer. Please return the revised version within the next 60 days. If you anticipate any delay in its return, we ask that you let us know the expected resubmission date by email at ploscompbiol@plos.org. Revised manuscripts received beyond 60 days may require evaluation and peer review similar to that applied to newly submitted manuscripts.

Sincerely,

Lyle J. Graham

Deputy Editor

PLOS Computational Biology

[LINK]

Reviewer's Responses to Questions

**Comments to the Authors:**

Reviewer #1: This is an interesting paper that provides a new method of predicting bumblebee odor discrimination behavior using the cosine distance measure of the vectorial representation of the chemicals. The author has shown that this method is capable of replicating the bumblebee behaviour experimentally too, and has identified a range of angles representative of generalizablity and discrimination. It has also been claimed that this method could help in faster screening of odor pollutants in the bumblebee ecology. Although, a similar work has been done in human odor space, wherein Snitz et al[1] showed that the odor similarity perceived by humans can be predicted by the cosine distance of the odor structures, a similar result in bees is interesting.

I have a few comments/questions:

1. At the outset, I would to like to request the author to provide page numbers and perhaps line numbers in the manuscript; it helps in pointing out the exact location being referred to.

2. Please define or at least give the full form of the abbreviations before their usage, such as CO (Contrasting odor), Brg- Bergamot and other oils in Table 1. It would make life easier for the readers who are not from a similar background.

3. The title ‘compound without borders (COB)’ although catchy does not seem to justify the context or meaning. Is the author talking about the compounds or the descriptors, or the method which is claimed to be universal in nature? The idea of many to many relationships between compounds and receptors is well known and accepted so, what is so novel about the idea of COB? I think the title should justify the contribution.

4. There are 66 claimed descriptor sets of compounds sufficient to be described as vectors which can be used for distance calculation, how were these specific set and numbers reached at? Please elaborate more on their importance. A simple linear combination of the features is interesting, is there any evidence in this research pointing towards linear relationship of the features governing the neuronal response?

5. What is the meaning of the sentence ‘odor space is constant’? If the odor space is constant, then what is the meaning of different odor in this context?

6. It would be informative to at least name the 5 additional types of associative odor used for comparison task described in Table S2.

7. The calculation of interval sets in identifying the theoretical distribution should be elaborated more, because it forms an important part of the contribution i.e. how and why these ranges were reached at? Moreover, I think the author should also provide the uncertainty involved in reporting expected number of Correct (C), Incorrect (IC) and No Choice (NC), because a point estimation of these values does not convey the real picture. What was the underlying assumption in C, IC and NR i.e. apriori was the assumption that C, IC and NC will be in equal numbers? Was the assumption of posterior a multinomial distribution with the C, IC and NC being independent and P(C) + P(IC) + P(NC) = 1. If yes, then one of the methods could be taking the parameters of the multinomial distribution from a Diritchlet prior.

8. Fig 4 description says subthreshold CO and suprathreshold angles to be 25.1 and 57 respectively but the figure has 24.7 and 56.7, please verify.

9. What is the range of COB angles in bumblebee odor space? Is there a limit to the COB angle in greater than 30 range? Is the 30 degree a hard threshold?

10. The claim of ease of method over PCA seems a bit stretched, because in this method too one has to obtain a 66 dimensional vector first by using GC-MS and then obtain the cosine distance for any inference. PCA is just another statistical step which is to be conducted.

Reference

1. Snitz, K., Yablonka, A., Weiss, T., Frumin, I., Khan, R.M. and Sobel, N., 2013. Predicting odor perceptual similarity from odor structure. PLoS computational biology, 9(9), p.e1003184.

Reviewer #2: In MS#COMPBIOL-D-19-01093, by Sprayberry, the author examines an important question in olfactory biology: how do insects process and discriminate complex scents? The study uses a combination of chemical analyses, behavioral (fmPER) and computational methods, the author seeks to examine the probability that an insect can identify different complex scents. I am enthusiastic about the framework of the manuscript, but my enthusiasm is somewhat lessened by the limited scope of the data used in the study.

1. I would appreciate more data analysis in the author's approach. For instance, there has been a tremendous growth in the number of studies showing that floral scents can often cluster based on the primary pollinator 'functional group' (borrowing Fenster et al. 2004 terminology) visiting a floral species. How would using these different datasets change the author's results? Data from Knudsen (1991), Rachel Levin (2001, 2005), Andreas Jurgens and many others could be used to examine whether scent chemistry is predictive of the pollinator group, and how slight changes to the scent alters this link.

2. How well does the author's analysis compare to traditional multivariate analyses (NMDS) or machine learning (ML) approaches (random forest)? Although PCA was a traditional analysis early on, more recent methods use ML approaches. Related to the data input, although the author is using the large peaks in the GCMS chromatogram, other smaller peaks can also play an important role in insect perception. This is also reflected in the variation of the scent. The author uses essential oils in the analysis and experiments, but often natural scent can be extremely variable individual by individual. How would this change the author's analysis and results?

3. The author also brings up an important point about how scent pollution may alter these scent profiles. But there is a gap in these hypotheses and the analyses performed. For instance, ozone and NO3 will modify the scent composition, but the author does not address how such degradation may be reflected in the analysis. If the author adds an agrochemical pollutant, how does that modify the scent differential, and influence behavior using FMPER?

4. Moreover, with respect to the insect, datasets exist for functional characterization of the odor constituents activating the olfactory sensory neurons (Drosophila, Anopheles, Tephritids, others...)(see Biasazin et al., 2018) - can the author use related dataset(s) to match the chemistry with the insect perception?

5. The author brings up many different topics in the Introduction and Discussion (scent discrimination and composition, encoding of the scent, scent pollutants, etc... From the readers perspective, it would help to focus the Intro on a few testable hypotheses that relate to the experiments in the Results: what is the relationship between complex scents and insect perception?

**Have all data underlying the figures and results presented in the manuscript been provided?**

Reviewer #1: Yes

Reviewer #2: No: The author mentions other datasets, but this is not included in the manuscript.

PLOS authors have the option to publish the peer review history of their article (what does this mean?). If published, this will include your full peer review and any attached files.

Reviewer #1: No

Reviewer #2: No

---

## [Decision Letter · Decision Letter 1]

26 Nov 2019

Dear Dr Sprayberry,

Thank you very much for submitting your manuscript 'Compounds without borders: a mechanism for quantifying complex odors and responses to scent-pollution in bumblebees' for review by PLOS Computational Biology. Your manuscript has been fully evaluated by the PLOS Computational Biology editorial team and in this case also by independent peer reviewers. The reviewers appreciated the attention to an important problem, but raised some substantial concerns about the manuscript as it currently stands. While your manuscript cannot be accepted in its present form, we are willing to consider a revised version in which the issues raised by the reviewers have been adequately addressed. We cannot, of course, promise publication at that time.

We are sorry that we cannot be more positive about your manuscript at this stage, but if you have any concerns or questions, please do not hesitate to contact us. That being said, I think that you should be able to sufficiently address the reviewers' concerns.

Sincerely,

Lyle Graham

Deputy Editor

PLOS Computational Biology

[LINK]

Reviewer's Responses to Questions

**Comments to the Authors:**

Reviewer #1: Review

Compounds without borders: a novel paradigm for quantifying complex odors and responses to

scent-pollution in bumblebees.

The line numbers specified here point to the modified new manuscript.

The author has made some specific changes to the manuscript and I have a few comments/suggestions on the new version.

1. In reference to the statements.

‘...then used a probability model to determine the level of uncertainty in this distribution, where individual silica bees are assigned a correct, incorrect, or no-choice response based

on the relationship between a random number from 0-1 and the expected response distribution. I ran this model with 177 bees (to match the above sample size) 17 times (to

match the number of experiments conducted in this study) to estimate the a standard deviation for each response category (SD C =4.1, SD I 2.9, SD NC =3.4 ). I edited the manuscript to reflect this updated approach (lines 697-705).’

I am afraid, the method described here is still not clear, for example which probability model was used to assign correct, incorrect, no-choice response. I understand it has been said to depend upon the relationship between a random number from 0-1 and the expected distribution but, does it depend upon the proportion of the observed distribution? To be exact, what was the apriori assumption in assigning the random number to a particular class of C, IC, NC? I think the usage of terms expected and observed is confusing. The expected in general comes out of the theoretical distribution and observed is what is found from actual experiments.

Moreover, the number of runs to get the SDs are too less (they should be run at least 100 times to get a proper distribution i.e. it is not sufficient to match the number of experiments done practically).

2. One more thing that is concerning is the less number of examples of chemicals between 20-29 degrees range (not the samples or replicates). Any claim in this range has to be verified with more data or stated clearly to be inconclusive.

3. Line 65- ‘Scent contaminants cause an additive-modifiction of scent blends.’.. please correct the spelling of modification.

Does the author have some evidence to back this claim as this is an interesting and strong statement, the change of bumblebee behaviour says nothing about additive/multiplicative or any such modification? It only says there is a change. There has to be some evidence in physico-chemical and neural space to claim this.

4. Line 79-81, ‘The principle goal of this study was to develop an odor-space with quantitative independent-axes that could be used for both descriptive and predictive analysis of bumblebee behavior’.

If the author claims to have found out independent axes in bumblebee odor space then it has has to be verified mathematically and analytically, what are these axes, how are vectors related in this space, is this a euclidean space? Because this is a long and often elusive question in the odor space research i.e. to find independent axes. I think the main contribution of this paper is to provide a new method of predicting bumblebee odor discrimination behaviour using the cosine distance measure of the vectorial representation of the chemicals. The claim of independent axes in my humble opinion should be avoided.

5. I am still not convinced with the term compound without border (CWB) and its use as title. I will let the author decide the relevance of use of this term as title.

6. Line 174- ‘..discriminate between an associative odor (AO) and unscented mineral oil (Fig. 2C)..’

there is no figure 2C. Please verify.

7. Line 198- ‘..LoV-associated bumblebees and 55.6% for JB-associated (Fig. 4).’

Is it fig 4 or fig 3 ?

8. Fig. 5 X-axis should have more markers in X-axis to depict the angles more clearly.

9. Line 242 - ‘.. and a suprathreshold CO ((theta= 57, Crd vs Brg...’

The caption says 57 but the figure has theta = 45. please verify.

10. Line 351- ‘..in the alkane and cyclic dimensions than any of the other odors (Fig S?).’

Please check which figure is being referred to.

Reviewer #2: In the revised manuscript, the author carefully addresses the reviewer comments. I think the revised manuscript is almost ready for prime time. I only have a few minor comments that should be easily addressed.

1. Introduction. The Intro, as rendered in the pdf, is difficult to read, perhaps because it is a single, long paragraph, with many distinct topics. For readability, I would encourage the author to separate the Introduction into distinct paragraphs, each with a topic sentence and concluding sentence to provide continuity between sections while addressing the different topics. For example:

A. The importance of floral advertisements and scent as an important cue that acts as a long-distance attractant.

B. How floral scents are subject to contaminants (anthropogenic and natural).

C. Statistical methods do not adequately enable predictive ability to characterize these dynamics.

D. Insect olfactory processing, CWB, and the discrete hypotheses (or questions) addressed in this manuscript.

2. For clarity and avoiding anthropomorphic terms, I would suggest changing a few phrases in the manuscript:

A. Fig. 1 Legend (Lines 133-135): Change "species" to "type" to avoid confusion about the bee or moth species used in the manuscript.

B. Line 108: Change "mechanism" to "analysis".

C. Change heading title on Line 247 to "Scent contamination reduces odor discrimination". We're not sure that the scent is unrecognizable to the bee (could be "recognizable" depending on context, bioassay, etc..).

3. Line 351: Insert the supplemental figure number.

**Have all data underlying the figures and results presented in the manuscript been provided?**

Reviewer #1: Yes

Reviewer #2: Yes

PLOS authors have the option to publish the peer review history of their article (what does this mean?). If published, this will include your full peer review and any attached files.

Reviewer #1: No

Reviewer #2: No

---

## [Decision Letter · Decision Letter 2]

2 Mar 2020

Dear Dr. Sprayberry,

We are very pleased to inform you that your manuscript 'Compounds without borders: a mechanism for quantifying complex odors and responses to scent-pollution in bumblebees' has been provisionally accepted for publication in PLOS Computational Biology.

Best regards,

Lyle J. Graham

Deputy Editor

PLOS Computational Biology

Reviewer's Responses to Questions

**Comments to the Authors:**

Reviewer #1: Thank you for addressing the changes and questions. I congratulate the author for this interesting work.

Reviewer #2: The author has addressed my comments

**Have all data underlying the figures and results presented in the manuscript been provided?**

Reviewer #1: Yes

Reviewer #2: Yes

PLOS authors have the option to publish the peer review history of their article (what does this mean?). If published, this will include your full peer review and any attached files.

Reviewer #1: No

Reviewer #2: No

---

## [Editor Report · Acceptance letter]

30 Mar 2020

PCOMPBIOL-D-19-01093R2 

Compounds without borders: a mechanism for quantifying complex odors and responses to scent-pollution in bumblebees

Dear Dr Sprayberry,

I am pleased to inform you that your manuscript has been formally accepted for publication in PLOS Computational Biology. Your manuscript is now with our production department and you will be notified of the publication date in due course.

With kind regards,

Laura Mallard
